# Body Composition Change, Unhealthy Lifestyles and Steroid Treatment as Predictor of Metabolic Risk in Non-Hodgkin’s Lymphoma Survivors

**DOI:** 10.3390/jpm11030215

**Published:** 2021-03-16

**Authors:** A. Daniele, A. Guarini, S. De Summa, M. Dellino, G. Lerario, S. Ciavarella, P. Ditonno, A. V. Paradiso, R. Divella, P. Casamassima, E. Savino, M. D. Carbonara, C. Minoia

**Affiliations:** 1Experimental Oncology and Biobank Management Unit—IRCCS Istituto Tumori “Giovanni Paolo II”, 70124 Bari, Italy; a.paradiso@oncologico.bari.it (A.V.P.); rosadive@inwind.it (R.D.); 2Hematology Unit—IRCCS Istituto Tumori “Giovanni Paolo II”, 70124 Bari, Italy; attilioguarini@oncologicco.bari.it (A.G.); giovanna.lerario@libero.it (G.L.); sabinociavarella@yahoo.it (S.C.); paolo.ditonno@fastwebnet.it (P.D.); carlaminoia@libero.it (C.M.); 3Molecular Diagnostics and Pharmacogenetics Unit—IRCCS Istituto Tumori “Giovanni Paolo II”, 70124 Bari, Italy; desumma.simona@gmail.com; 4Gynecologic Oncology Unit—IRCCS Istituto Tumori “Giovanni Paolo II”, 70124 Bari, Italy; miriamdellino@hotmail.it; 5Clinical Pathology Laboratory Unit—IRCCS Istituto Tumori “Giovanni Paolo II”, 70124 Bari, Italy; porzia.casamassima@oncologio.bari.it (P.C.); eufemiasavino@libero.it (E.S.); mdcarbonara@yahoo.it (M.D.C.)

**Keywords:** non-Hodgkin’s lymphoma survivors, metabolic syndrome, body composition change, unhealthy lifestyle, steroid use, sarcopenia, diffuse large B-cell lymphoma

## Abstract

Unhealthy lifestyle, as sedentary, unbalanced diet, smoking, and body composition change are often observed in non-Hodgkin’s lymphoma (NHL) survivors, and could be determinant for the onset of cancer treatment-induced metabolic syndrome (CTIMetS), including abdominal obesity, sarcopenia, and insulin resistance. The aim of this study was to assess whether changes in body composition, unhealthy lifestyles and types of anti-cancer treatment could increase the risk of metabolic syndrome (MetSyn) and sarcopenia in long-term NHL survivors. We enrolled 60 consecutive NHL patients in continuous remission for at least 3 years. Nutritional status was assessed by anthropometry-plicometry, and a questionnaire concerning lifestyles and eating habits was administered. More than 60% of survivors exhibited weight gain and a change in body composition, with an increased risk of MetSyn. Univariate analysis showed a significantly higher risk of metabolic disorder in patients treated with steroids, and in patients with unhealthy lifestyles. These data suggest that a nutritional intervention, associated with adequate physical activity and a healthier lifestyle, should be indicated early during the follow-up of lymphoma patients, in order to decrease the risk of MetSyn’s onset and correlated diseases in the long term.

## 1. Introduction

B-cell non-Hodgkin’s lymphomas (NHL) are a heterogeneous group of lymphoproliferative disorders originating from B-lymphocyte. They are the seventh most frequent neoplasm among both men and women, accounting for 4–5% of new cancer cases. Among them, diffuse large-B-cell lymphoma (DLBCL) is the most common histotype, accounting for about 30% of all cases [1,2]. The median age at diagnosis is 70 years, but patients at a younger age could also be affected. The cure rate for DLBCL is steadily increasing over time, with 5-year overall survival rate ranging from about 40 to 90%, according to prognostic groups. The numbers of lymphoma survivors have been increasing over recent decades, and represent a population at risk for several late-onset toxicities [3,4].

With many lymphoma patients surviving years after treatment, it is important to understand the long-term effects of cancer treatment, including its effects on weight and body composition [5]. The main long-term effects of chemotherapy, radiotherapy and autologous stem cell transplant (ASCT) in NHL survivors are cardiotoxicity and secondary neoplasms, but other toxicities, such as fatigue, anxiety and depression, infertility, and endocrine-metabolic disorders, could also occur, so that a multi-disciplinary approach to these patients is often needed [6]. In several cohorts of NHL survivors, a relevant prevalence of individual and modifiable risk factors for late sequel has been reported [7,8,9,10]. In particular, they could present unhealthy lifestyles (inactivity, unbalanced diet, smoking habit) and overweight, which could be the bases for the development of cancer treatment-induced metabolic syndrome (CTIMetS) and cardiovascular diseases.

The causes, complications, diagnosis and treatment are similar to those of obesity. Visceral adiposity is a key component of the Metabolic Syndrome (MetSyn), and has been associated with higher blood levels of metabolic parameters, such as glycaemia, total cholesterol and triglycerides [11,12,13], hypertension [14,15], and increased risk of coronary artery and type 2 diabetes [16,17]. On the other hand, CTIMetS and sarcopenia, or low muscle mass, have been associated with poorer functional outcomes and increased sedentary habits in survivors [18]. The most accredited diagnostic criteria for MetSyn have been established by the National Cholesterol Education Program Adult Treatment Panel III [19]. Central obesity plays a key role in MetSyn onset, reflecting the hypothesis that visceral adipose tissue and subcutaneous fat might directly lead to MetSyn, because of their hyperlipolytic state and the contributions of excess free fatty acid to insulin resistance; besides a reduction in muscle mass, known as sarcopenia, an increase in fat mass is one of the most striking and consistent changes associated with obesity. Sarcopenia is considered a muscle disease (muscle failure) presenting with a reduction in both muscle quantity and quality. Adipose tissue is increasingly considered an endocrine organ that in obese people releases an excess of pro-inflammatory cytokines, such us interleukin–6 (IL-6) and tumor necrosis factor-α (TNF- α) [20]. Dyslipidemic features, including hypertriglyceridemia, higher low density lipoprotein coleterol (LDL– C) serum levels and low high-density lipoprotein cholesterol (HDL- C) levels, have been described in cancer survivors [21,22,23]; the findings from many controlled and well-designed studies have demonstrated that cancer survivors are more dyslipidemic than a control population.

The features of dyslipidemia are closely related to cardiovascular risk. A greater risk of cardiovascular events has been observed when patients received cardiotoxic drugs, mainly anthracycline-containing regimens, mediastinal radiotherapy and ASCT [24,25]. On the other hand, lifestyle factors are well known to be major risk factors for MetSyn, with a role widely demonstrated in the literature.

Taking into account these aspects, the aim of this study was to evaluate whether unhealthy lifestyles, overweight, and variations in body composition might be risk factors for the onset of cancer CTIMetS, as a long-term sequela in NHL survivors after cancer treatment.

## 2. Patients and Methods

### 2.1. Study Design

This is a cohort study including consecutive NHL patients in remission for at least 3 years and in current follow-up. The patients were prospectively evaluated by the multidisciplinary team dedicated to lymphoma survivors’ care and prevention, as part of a monocentric cohort study conducted from November 2016 to September 2020 at the IRCCS Istituto Tumori “Giovanni Paolo II” in Bari, for the project “Centre for Disease Control Program 2014 (CCM2014)”, financed by the Italian Ministry of Health. The diagnosis of NHL was made according to the WHO classification. Clinical data on diagnosis, treatment approaches, comorbidities, medications and symptoms were obtained thought medical records and hematologic follow-up visits, and were collected in a dedicated database. Each patient underwent to a nutritional assessment by anthropometry and plicometry measurement. Contextually, questionnaire about lifestyles, food habits, therapies carried out and in progress, allergies or intolerances, and sleep disorders was administered. Unbalanced diet was defined as a diet rich in simple sugars or complex carbohydrates and hydrogenated fats, and low in fiber or legumes and whole grains. The study was conducted according to the declaration of Helsinki. The study was approved by the Ethical committee, and each patient signed their informed consent

### 2.2. Body Composition Assessmen

The weight and height of the patients were measured to scale with an altimeter (Millenium 3 DAVI and CIA—Barcelona, Spain). The main circumferences of the body, including wrists, arms, chest, waist, hips, and mid-thigh, were measured with tape for body circumferences (SECA 1017526—Amburgo, Germany) [26], while the percentage of fat mass (FT) and lean mass (FFM) was assessed using a FAT-1 plicometer (GIMA- Gessate (MI), Italy), measuring seven skinfolds according to the Durnin–Womerslay criteria. [27].

MetSyn is defined, according to the National Cholesterol Education Program Adult Treatment Panel III, as the presence of at least three of the following criteria: (1) waist circumference (due to excess abdominal fat) > 88 cm (cm) for females and >102 cm (cm) for males; (2) high blood pressure (>130/85 millimeters of mercury (mmHg)); (3) dyslipidemia (triglyceride >150 milligram per deciliter (mg/dL)); (4) high-density lipoprotein cholesterol (HDL-C) (<40 mg/dL for male and <50 mg/dL for females); (5) impaired fasting glucose (>110 mg/dL) or insulin resistance.

The waist hip ratio (WHR), used in clinical practice to evaluate the distribution of body fat, was obtained from the ratio of the waist/hip circumferences. This index is affected not only by the increase in visceral fat, but also by the loss of muscle; obesity type status (android, gynoid or intermediate) was assessed with the WHR.

The World Health Organization’s (WHO) definitions of obesity (body mass index (BMI) > 30 kg/m^2^) and overweight (BMI 25–29.9 kg/m^2^) were used [28]. The basal energy expenditure (BEE) was calculated according to the Harris and Benedict equation (HBE) [29], while the ideal weight of each patient was calculated using the Lorenz equation, as described in our previous work.

### 2.3. Metabolic Parameters’ Measurement

Measurement of metabolic and inflammatory parameters was performed using COBAS c311/501 (Roche-Diagnostic Limited S.p.A Basel- Switzerland).

Dyslipidemic features, including hypertriglyceridemia, high low-density lipoprotein cholesterol (LDL-C), and low HDL-C levels, were evaluated. In particular, dyslipidemia IIa (hypercholesterolemia) and IIb (hypercholesterolemia + hypertriglyceridemia) were considered according to the classification of Fredrickson’s phenotypes, derived from the World Health Organization (WHO).

### 2.4. Statistical Analysis

Univariate and multivariate logistic regression were performed through a generalized linear model, using binomial family and logit link. The “MASS” R package was used in the R studio Version 1.1.447 [30]. The results are reported as hazard ratio and 95% confidence interval, and they were considered significant when the *p*-value was > 0.05. The mean comparison of serum and anthropometric parameters was evaluated using the analysis of variance (ANOVA). A *p*-value ≤ 0.05 was considered statistically significant.

## 3. Results

In total, 60 NHL survivors were enrolled, 34 were women and 26 men, with a median age of 46.5 (range 22–82), The vast majority was homogeneously represented by DLBCL (*n* = 57, 95%), and the remaining by follicular lymphoma (FL) grade III (*n* = 3, 5%). They presented a median of 8 years since remission of the disease (range, 4–48).

All patients (100%) had been treated, as the front-line treatment, with R-CHOP (rituximab, cyclophosphamide, vincristine, doxorubicin, prednisone), CHOP, or CHOP-like regimes, which included high-dose steroids. A total of 54 (90%) also received radiotherapy. Nine patients (15%) had received salvage chemotherapy followed by ASCT. 

Table 1 shows the disease, anthropometric characteristics, and lifestyles of all the patients enrolled.

Most patients presented unhealthy lifestyles: 35/60 (58%) presented an unbalanced diet, 38/60 (63%) physical inactivity, and 47/60 (78%) smoking habits.

The results of this study showed that 60% of the patients suffered from MetSyn, with a higher waist circumference value than patients without MetSyn in women and men (98 ± 17 vs. 84 ± 11; *p* = 0.001; 104 ± 9.0 vs. 93 ± 8.1; *p* = 0.005). A statistically significant association was also observed in body composition change, in particular a decrease in FFM (fat-free mass) in men suffering from MetSyn with respect to those without MetSyn (67.9% and 71.7%, respectively; *p* = 0.002). Furthermore, a significant correlation was observed in serum levels of β-2microglobulin, as a pro-inflammatory marker, with higher values in patients with MetSyn compared to those without (1.08 ± 0.45 vs. 1.82 ± 2.0; *p* < 0.001). Data reported in Table 2.

Anthropometric and nutritional parameters were compared, stratifying patients via the occurrence of MetSyn (Table 3) and the presence of MetSyn, and dyslipidemia IIa and IIb was assessed in relation to the following: (i) anti-cancer treatment (use of high-dose steroids within chemotherapic regimen, ASCT) (ii) unhealthy lifestyles (smoking habit, poor physical activity and unbalanced diet); iii) hypertension, and food intolerance.

The univariate logistic regression highlighted a significantly high risk of MetSyn in survivors affected by hypertension (HR: 10.98; *p*-value: 0.000158). A statistical trend indicating a high risk of MetSyn in patients that underwent ASCT has also been observed (HR: 7.96, *p*-value: 0.06), as well as a higher risk of MetSyn onset, which was observed among subjects who had poor physical activity (HR: 0.32; *p*-value: 0.02) and those who followed an unbalanced diet (HR: 0.30; *p*-value: 0.02) (Table 3). The multivariate logistic regression for MetSyn highlighted a higher risk in patients that underwent ASCT and who had hypertension (HR: 49.52 and 35.42, respectively).

An increased risk of the onset of MetSyn was associated with the use of steroids administered in second-line chemotherapy and a smoking habit (HR: 6.80 and 2.67, respectively). Moreover, the presence of food intolerance reduced such a risk (HR: 0.10) (Figure 1), probably due to limitation in the diet, as observed for multivariate analysis regarding dyslipidemia IIa (Figure 2). Regarding dyslipidemia IIb, hypertension was found to be related to a higher risk (HR: 8.43), as a statistical trend. The presence of sarcopenia is responsible for a significantly lower risk of dyslipidemia IIb. (Figure 3)

## 4. Discussion

Diabetes, hypertension and obesity are increasingly common clinical conditions in the population due to incorrect habits and lifestyles. This explains the high incidence of MetSyn. It is estimated that in Italy, about 25% of the general population suffers from it, and genetic predisposition and environmental conditions, such as a sedentary lifestyle and weight gain, play a fundamental role in its onset. However, other factors also affect its development, such as age, smoking, diets rich in carbohydrates, alcohol consumption, and low physical activity.

Metabolic disorders also represent a consistently emerging long-term sequela of anti-cancer treatment in the growing population of cancer survivors. Among them, cancer treatment-induced metabolic syndrome (CTIMetS) differs from the MetSyn reported in the general population as regards pathogenesis and etiology, but the prevention methods are probably similar. Chemotherapy appears to contribute to the patho-physiology of CTIMetS, mainly through gonadal toxicity, inducing decreased levels of estrogen and testosterone, which are associated with central obesity, dyslipidemia and insulin resistance [31,32].

In long-term lymphoma survivors, treatment with chemotherapy is associated with a higher risk of weight gain and the onset of MetSyn. Changes in adipose tissue and lean mass are two alterations induced by anti-cancer treatment, often called sarcopenic obesity. On the other hand, physical inactivity, unbalanced nutrition and unhealthy lifestyles continue after cancer treatment, and may also be responsible for asthenia and FFM reduction, which in turn induce a decrease in insulin-stimulated glucose uptake. Damage to the gastrointestinal tract and liver may also impair insulin sensitivity; gut motility is impaired not only by cancer treatment, but also via dysbiosis of the intestinal microbiome by effect of diet restriction and antibiotics, which are also commonly used during cancer treatment [33,34,35]. 

Several studies in the literature have highlighted the association between MetSyn, obesity, and the poorer prognosis of cancer. Several authors have also focused their attention on those who have recovered from cancer, who are, however, in relation to the therapies received, at higher risk of developing metabolic disorders [3,21,36,37,38]. What emerges from these works is that the causes of the onset of MetSyn in cancer survivors differ from those in the healthy population; in effect, local treatments (surgery and radiotherapy) and systemic cancer therapy (chemotherapy and hormone therapy) can damage the functionality of the endocrine system, and this would represent the first step in the onset of metabolic disorders.

In sarcopenic obesity, sarcopenia and obesity’s patho-physiologies are strongly interconnected. The scientific evidence suggests that the body percentage variability of FFM and FM are strongly dependent: for any increase in body fat, a parallel change in FFM occurs, corresponding to approximately 25% [39]. The loss of muscle mass and function can also be favored by obesity as an independent risk factor, due to the alterations related to it, such as chronic low-grade inflammation, increased oxidative stress, insulin resistance (with decreases in the anabolic capacity of the striated muscles, in addition to the worsening of the carbohydrate profile), a sedentary lifestyle, and a higher incidence of chronic diseases that can have a negative impact on muscle catabolism [40]. Chronic low-grade inflammation, present in obesity and promoted by it, causes a decrease in muscle anabolic function, mediated by changes in the production of factors, such as TNF, IL-6, leptin and GH [41].

Obesity also favors the accumulation of adipose tissue in the ectopic area of the liver, bone marrow and skeletal muscles, determining the processes of lipotoxicity and inflammation. Intramuscularly, fat accumulates mainly between the different muscle groups, delimited by bands. The intramuscular adipose tissue (IMAT) can promote alterations in the differentiation of progenitor mesenchymal cells, compromising the regenerative potential of the muscle and stimulating the establishment of fibrosis processes. Muscle adipocytes also secrete paracrine hormones and pro-inflammatory cytokines that promote a feedforward cycle by producing intramyocellular lipids. This lipotoxicity impairs the contractility of muscle fibers and interferes with muscle protein synthesis, exacerbating sarcopenia [42,43]. Lipid deposition can also occur in spaces previously occupied by muscles, impairing the growth of new muscle tissue. 

The major risk factors for sarcopenic obesity could be sedentary lifestyle and disability, poor nutrition, frequent weight changes, and the yo-yo effect, which induce muscle mass depletion, chronic low-grade inflammation and endocrine-metabolic diseases, such as type 2 diabetes mellitus, insulin resistance, hypogonadism and hypercortisolism.

Beyond the long-term damage caused by anti-cancer treatments to body composition, many unhealthy lifestyles, including unbalanced diet, a smoking habit and poor physical activity, could play a significant role in determining an individual’s risk of developming MetSyn [7]. The present study is the first analyzing whether unhealthy lifestyles, as recognized by current guidelines on survivorship, can have an impact on the onset of MetSyn in long-term NHL survivors, mainly represented by DLBCL. Our results showed that about 60% of NHL survivors were diagnosed with MetSyn and excess malnutrition, with a high metabolic risk in 60% of cases. The diagnosis of MetSyn was present in both men and women. No correlation with age was found. According to the received anti-cancer treatment, it has to be highlighted that those with both high-dose steroid treatment, administered within first- or second-line therapy, and intensification with ASCT, were at a higher risk of developing MetSyn. According to individual factors, hypertension was found to correlate with the onset of MetSyn and dyslipidemia IIb, contributing to a further excess of risk for cardiovascular disease. Considering unhealthy lifestyles, inactivity, unbalanced diets and smoking habits significantly increased the risk of MetSyn. The evaluation of the questionnaire administered to each patient highlighted that most of the subjects in the post-treatment period did not follow an adequate diet, with a poor distribution of meals and most of the caloric intake in the evening hours, associated with low activity physical, sometimes due to asthenia and joint pain over time, as well as smoking habit. The authors also found body composition changes in the long-term follow-up period, and we additionally observed a significant loss of lean mass (sarcopenia) in the MetSyn group. This aspect reflects a substantial body composition change, which is often reflected in chronic fatigue and poor quality of life in lymphoma survivors, as well in the whole population of cancer survivors [44]. Sarcopenia is often underestimated, due to its being masked by weight gain. The process of sarcopenia could be inscribed in a more complex chronic inflammatory process, as demonstrated by the higher levels of β-2 microglobulin.

Although current guidelines recommend the strict control of unhealthy lifestyles among cancer long-term survivors in general (NCCN 2020), there are no specific indications for the prevention and treatment of MetSyn in this growing population. Nutritional assessment and intervention in these patients should be ensured early by promoting comprehensive care activities, which provide all the correct information on the lifestyle to be undertaken after treatments and in the follow-up. To this end, an oncological nutrition assessment is detrimental.

## 5. Conclusions

The general guidelines indicate that a balanced diet and a correct lifestyle have been shown to be the first steps towards preventing MetSyn. Therefore, it would be desirable for long-term lymphoma survivors, especially those at high risk of metabolic disorders and sarcopenia, to implement appropriate and early interventions into their lifestyle and nutrition in order to prevent the onset of MetSyn, and consequently prevent obesity, sarcopenia and cardiovascular disease. 

## Figures and Tables

**Figure 1 jpm-11-00215-f001:**
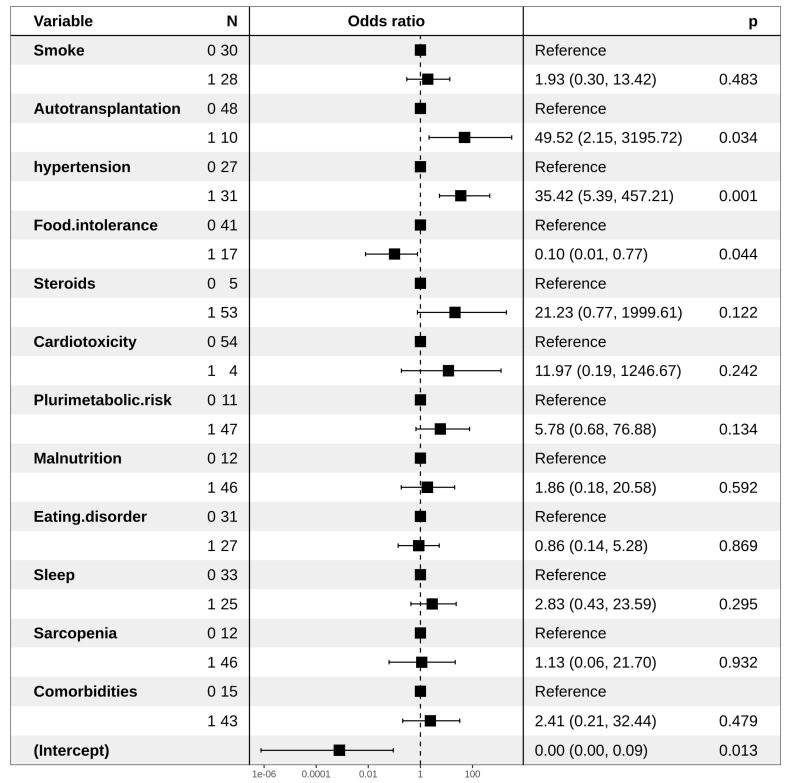
Forest plot representing multivariate logistic regression for a MetSyn.

**Figure 2 jpm-11-00215-f002:**
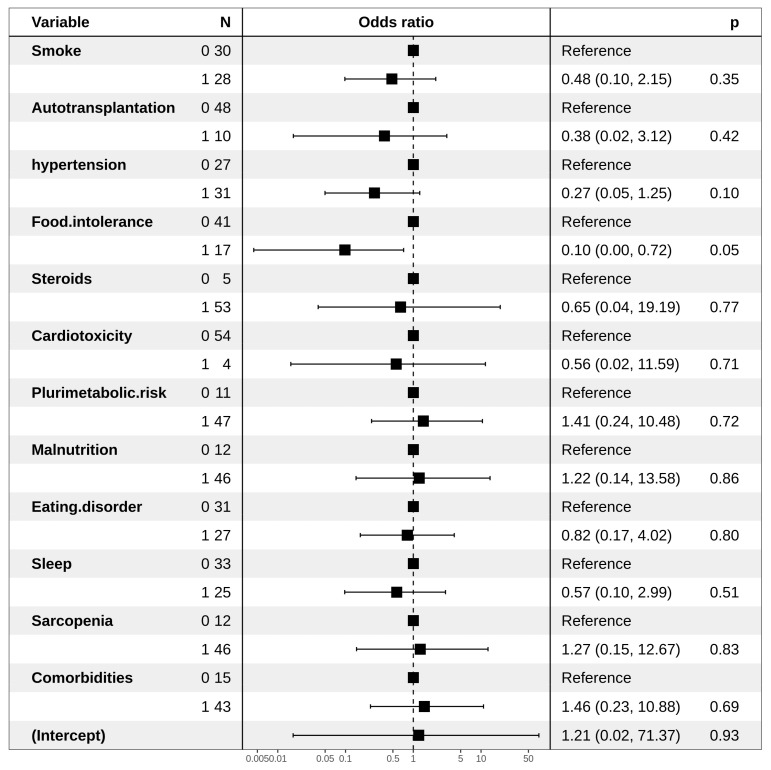
Forest plot representing multivariate logistic regression for dyslipidemia IIa.

**Figure 3 jpm-11-00215-f003:**
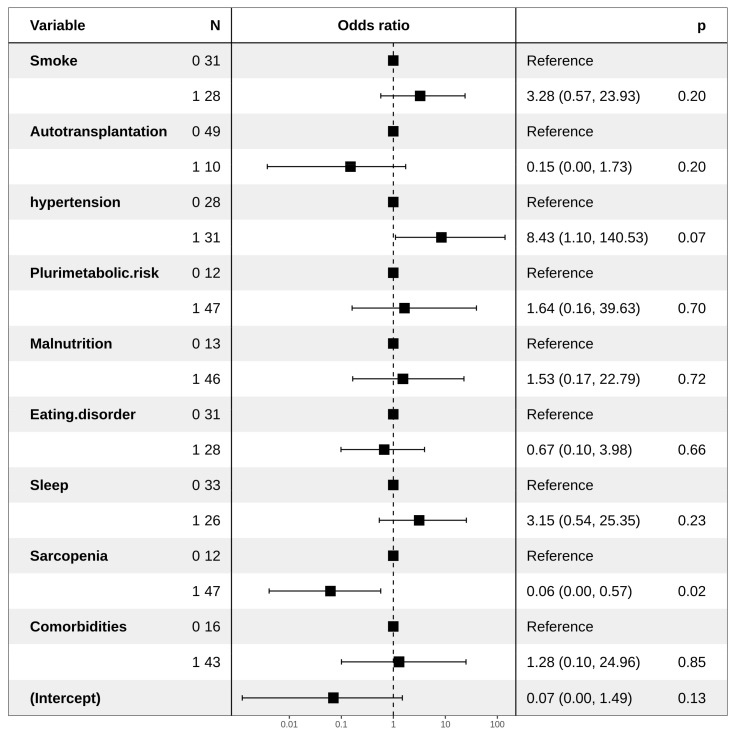
Forest plot representing multivariate logistic regression for dyslipidemia IIb.

**Table 1 jpm-11-00215-t001:** Baseline clinical- anthropometric and lifestyle characteristic of 60 NHL survivors.

Years	(Range)	
Male	(25–82)	
Female	(24–76)	
Hystotypes	n°	%
DLBCL	57	95
FL	3	5
Treatment	n°	%
Chemotherapy	60	100
Autologous transplantation	9	15
Radiotherapy	54	90
High-dose steroids	n°	%
Yes	50	83
No	10	17
Hypertension	n°	%
Yes (BP > 130/85 mmHg)	32	53
No	28	47
Weight Status	n°	%
Obesity/Overweight	43	72
Normal weight	8	13
Underweight	9	15
Biotype	n°	%
Ginoid	27	45
Android	17	28
Intermediate	16	27
Metabolic syndrome	n°	%
Yes	36	60
No	24	40
Excess malnutrition	n°	%
Yes	36	60
No	24	40
Unbalanced Diet	n°	%
Yes	35	58
No	25	42
Metabolic risk	n°	%
High	32	53
Low	28	47
Sarcopenia	n°	%
Yes	38	63
No	22	37
Smoke	n°	%
Smoker	15	25
Nonsmoker	13	22
Ex-smoker	32	53
Physical Activity	n°	%
Yes	22	37
No	38	63

DLBCL, diffuse large B-cell lymphoma; FL, follicular lymphoma; BP, blood pressure.

**Table 2 jpm-11-00215-t002:** Mean value comparison between anthropometric and metabolic parameters in non-Hodgkin’s lymphomas (NHL) patients with or without metabolic syndrome.

	DLBCL (*N* = 60)		
	MetSyn/yes (*N* = 36)	MetSyn/no (*N* = 24)	*p*-Value
Anthropometry			
Weight (Kg)	76.3 ± 14.0	73.0 ± 13.0	n.s
Height (cm)	169.5 ± 2.5	166.5 ± 6.7	n.s
BMI (kg/h^2^)	27.5 ± 4.5	26.6 ± 4.2	n.s
Waist circumference			
Women	98.0 ± 17.0	84.0 ± 11.0	0.001
Men	104.0 ± 9.0	93.3 ± 8.1	0.005
Percentage body fat			
Women	39.5 ± 5.0	38.4 ± 2.6	ns
Men	32.1 ± 4.2	28.2 ± 5.0	0.04
Percentage fat-free mass			
Women	60.6 ± 4.0	61.5 ± 2.5	0.45
Men	67.9 ± 4.1	71.7 ± 5.0	0.002
TDEE	2113 ± 309	2185 ± 275	ns
Inflammatory markers			
CRP (mg/dL)	2.10 ± 2.02	1.90 ± 1.01	n.s
β2-microglobulin (mg/L)	1.08 ± 0.45	1.82 ± 0.45	0.001
Metabolic parameters			
Glycemia (mg/dL)	101.5 ± 17.0	99.3 ± 7.2	n.s
Total cholesterol (mg/dL)	201.0 ± 30.8	199.0 ± 21.0	n.s
HDL-cholesterol (mg/dL)	52.4 ± 11.9	53.0 ± 11.0	n.s
Triglycerides(mg/dL)	133.2 ± 12	130.3 ± 6.0	n.s
Albumin (%)	55.5 ± 5.3	56.1 ± 4.7	n.s
TFR	250.5 ± 35.0	270.7 ± 31.0	n.s
HCT	41.7 ± 3.8	43.5 ± 2.5	n.s

MetSyn, Metabolic syndrome; DLBC, diffuse large B-cell lymphoma; BMI, Body Mass Index; TDEE, Total Daily Energy Expenditure; CRP, C-Reactive Protein); HDL- C, High Density lipoprotein- Cholesterol; TFR, Transferrin; HCT, Hematocrit.

**Table 3 jpm-11-00215-t003:** Univariate logistic regressions, with metabolic syndrome, dyslipidemia IIa and dyslipidemia IIb as dependent variables (statistically significant results have been reported, including statistical trend).

	Odds Ratio (95% CI)	*p*-Value
MetSyn		
Smoke		
No	Ref	
Yes	2.67 (0.92–8.16)	ns
ASCT		
No	Ref	
Yes	7.96 (1.34–152.57)	ns
Hypertension		
No	Ref	
Yes	10.98 (3.38–41.74)	0.0001
Food Intolerance		
No	Ref	
Yes	0.35 (0.11–1.10)	ns
Steroids		
No	Ref	
Yes	6.80 (0.93–138.07)	0.096
Unbalanced Diet		
No	Ref	
Yes	0.30 (0.10–0.82)	0.02
Physical Activity		
No	Ref	
Yes	0.32 (0.11–0.89)	0.01
Dyslipidemia IIa		
Hypertension		
No	Ref	
Yes	0.27 (0.06–0.93)	0.05
Food intolerance		
No	Ref	
Yes	0.14 (0.007–0.80)	ns
Dyslipidemia IIb		
Hypertension		
No	Ref	
Yes	4.52 (1.01–32.04)	ns
Smoke		
No	Ref	
Ex-Smoker	7.17 (1.94–30.49)	0.003
Yes	4.09 (0.89–19.23)	ns

MetSyn, Metabolic Syndrome; ASCT, Autologous Stem Cell Transplant.

## Data Availability

The data presented in this study are available on request from the corresponding author. The data are not publicly available due to ethical.

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
