# Peer review of "Body Composition Change, Unhealthy Lifestyles and Steroid Treatment as Predictor of Metabolic Risk in Non-Hodgkin’s Lymphoma Survivors"

_jpm, 2021, doi:10.3390/jpm11030215_

Round 1
Reviewer 1 Report
Daniele and coauthors enrolled 60 DLBCL survivors and they first measured their anthropometry and plicometry. They then analyzed the blood samples for measurement of triglyceridae, glycaemic, HDL-cholesterol, total cholesterol, β-2 microglobulin, albumin, C-reactive protein. The authors found that treatment with chemotherapy is associated with increased risk of weight gain and developing MetSyn. Furthermore, physical inactivity, unbalanced nutrition, and unhealthy lifestyles often coexist during and after cancer treatment. Finally, they concluded that early interventions on lifestyles and nutrition of long-term lymphoma survivors would provide an effective way for preventing the onset of the MetSyn.
The data presented support the conclusion drawn, which is useful for others in the related research field .
Author Response
Thank you very much for your review and valuable comment
Reviewer 2 Report
First of all, I want to give my congratulations to authors for the work done, for such a long time of collecting data. Nevertheless, it seems to me, that is missing a very important data in all the design of the experiment: the correlation with normal, healthy or not so healthy people. Just the difference between the patients that survive to DLBCL and were diagnose with or without MetSyn it seems very raw. Also, some data are described as significant relevance, when in reality, when it is compared the both it seems to me, that is not true.
For example, in line 180 and 181:
"among these patients a significant correlation was observed between the mean waist circumference (cm), higher in women with MetSyn (94 ± 17) compared to women without MetSyn (84 ± 11; p <0.05)"
The SD of the women without MetSyn put the value very close, almost the same, of the women with MetSyn. From my perspective and data analyses, it seem, not so significant.
Also, the configuration of the tables is very confusing, and not format all in same way. It should be used some graphs, because are easy to read and analyse.
Author Response
see fil

Reviewer 3 Report
In their paper Daniele et al. analyzed the risk of developing metabolic syndrome (MetSyn) in DLBCL survivors. The results, despite low case numbers, are interesting. Nevetheless some major and minor issues should be resolved befeore the paper may be published. These are enliseted below:
- I did not find the definition of metabolic syndrome in the paper. A clear definitione should be given;
- What was the histological type on analyzed non-Hodgkin lymhphoma cases – were there only DLBCL cases oraz other histological types analyzed? Data are inconsitent (please check Table 1);
- How many patients had a previous diagnosis of MetSyn before treamtment initiation? Did these patients form any other concomitant diseases?
- Please provide the definition of sarcopenia baed on the anthopometry measurements;
- Please provide the definitione of dyslipidemia type A and type B. Both these parameters are described in the results section however the significance of this finding is scanty described in the discussion;
- Did the development of MetSyn in the analyzed patient group contributed to the development of cardio-vascular diseases?
- Beta-2-microglobulin is considered as an inflammatory marker in the text. Please provide the rationale in the context of presented study. The beta-2-mikroglobulin levels are stronlgy dependent on creatinine clearence. Please provide this additional results and normalyse beta-2-microglobilin levels to creatinine clearence.
- The text requires some minor typing corrections
Author Response
see file attach

Round 2
Reviewer 2 Report
After read again the paper and the explanations to my concerns when I reject the paper, it seems much more clear now.
Just a note:
The first time authors used the acronym - WHO - it is not described. After that, they should use just WHO. (chapter 2 - patients and methods)